# Texture Characteristics of Sea Buckthorn (*Hippophae rhamnoides*) Jelly for the Elderly Based on the Gelling Agent

**DOI:** 10.3390/foods11131892

**Published:** 2022-06-26

**Authors:** Dah-Sol Kim, Fumiko Iida

**Affiliations:** Department of Food and Nutrition, Japan Women’s University, Tokyo 112-8681, Japan; dskim115@naver.com

**Keywords:** sea buckthorn, antioxidant, antidiabetes, elder-friendly food, texture, gelling agent

## Abstract

The aim of this study was to identify the nutritional components of sea buckthorn berries and to evaluate the hardness control of the elderly with mastication difficulties using various types and concentrations of gelling agents in the preparation of sea buckthorn jelly. As a result, sea buckthorn berry comprised various bioactive nutrients, including minerals, essential fatty acids, and antioxidative and antidiabetic substances. In addition, jelly added with 3.01% guar gum, 5.74% xanthan gum, and 11.38% locust bean gum had a smooth hardness that could be chewed with the elderly’s tongue. Guar gum at 3.23~6.40%, 6.02~9.90% xanthan gum, and 12.42~27.00% locust bean gum showed soft hardness that can be chewed with gum. These results show that the gelling agent is suitable for the development of food for the elderly that meets Korean Industrial Standards, considering the mastication difficulty and dysphagia in the elderly.

## 1. Introduction

Sea buckthorn (*Pastinaca sativia* L.) is a deciduous spiny shrub between 2 and 4 m high, widely grown to temperate climate region of Asia and Europe [1]. The most significant part is berries, which juice can be obtained from them. That is why sea buckthorn berries are the main reason for their high marketability. Moreover, sea buckthorn berries comprise various bioactive nutrients, including vitamins, minerals, essential fatty acids, free amino acids, and other principal elements [2]. Sea buckthorn berries are one of the most influential sources of these nutritionally valuable components, and it is clear that they can be used as a nutritional source in the commercial food market for the elderly. However, sea buckthorn berries have tough skin and fiber, making it difficult for the elderly to chew easily. In addition, there are few studies on sea buckthorn berries, making it difficult to know how to use them properly and making it less accessible to older consumers. Thus, even though many sea buckthorn berries are cultivated in Korea, many are being abandoned, and farmers are suffering because of this.

Texture is important not only for food palatability but also for the safety of eating in recent aged society [3]. For older people, particularly those with weakened chewing abilities, the textural characteristics of food should be changed from rheological features to make foods friendly to this group. The texture modification of the elderly’ food is now one of the most significant assignments in Korean food industry. Thus, the Ministry of Food and Rural Affairs in Korea set quantified information on the chewable hardness of food for the elderly at three levels [4]. The first level of hardness of the Korean Industrial Standards (KS), which defined this as a level that can be chewed by the teeth, is from 55,000 to 500,000 N/m^2^. The second level, which is stated as a level that can be chewed by gum, is 22,000~50,000 N/m^2^. The third level is the level that can be chewed with the tongue, and the hardness is less than 2000 N/m^2^. In addition, an appropriate viscosity (less than 1500 mPa.s) standard was also presented for the elderly with dysphagia. In other words, it takes into account the high risk of non-viscous drinks such as water flowing into the wrong airway due to weakening motility of the esophageal muscles due to aging. That is to say, the esophagus moves so that the food can go down naturally when it comes over to the esophagus, and if the exercise of the esophagus does not fit the rhythm, it may cause choking. The elderly, who have reduced esophageal muscle activity due to aging, are likely to have this problem easily not only when eating solid food but also water. If this problem continues to be repeated, it can cause an esophageal movement disorder, which is diagnosed by the Korean medical community as dysphagia. Moreover, it is recommended that the elderly drink viscous liquids as a precaution against dysphagia. This is why KS has prepared proper viscosity standards as well as hardness for the elderly. Reflecting these established standards, some studies have been conducted on texture properties suitable for the KS according to cooking method, enzyme, and other additives. Those studies indicated that the modification of texture could improve the balanced nutrient intake for the elderly [5]. However, there is almost no information about the applied sea buckthorn to food for the elderly. Therefore, this study was conducted to develop sea buckthorn jelly for the elderly suitable for the KS according to various gelling agents.

Here, guar gum, xanthan gum, and locust bean gum were used as gelling agents in this study, and the reasons are as follows. Guar gum is a galactomannan polysaccharide extracted from guar beans that has thickening and stabilizing properties useful in food. It has been shown to reduce serum cholesterol and lower blood glucose levels. In addition to guar gum’s effects on viscosity, its high ability to flow or deform provides it with favorable rheological properties. Xanthan gum is also a polysaccharide with many industrial uses, including as a common food additive. It is an effective thickening agent, emulsifier, and stabilizer that prevents ingredients from separating. It is also a preferred method of thickening liquids for those with swallowing disorders, since it does not change the color or flavor of foods or beverages at typical use levels. Locust bean gum is a galactomannan vegetable gum extracted from the seeds of the carob tree and used as a gelling agent in food technology. It is also used in many foods to thicken textiles. Considering these advantages, it was judged that guar gum, xanthan gum, and locust bean gum could provide meaningful development in the hardness of elder-friendly foods to be developed in this study. Therefore, in this study, the texture characteristics of sea buckthorn jelly for the elderly using a gelling agent to be suitable for KS were analyzed to confirm its applicability as an elder-friendly food.

## 2. Materials and Methods

### 2.1. Sample Preparation

Sea buckthorn berries cultivated in Hwacheon-gun (Gangwon-do, South Korea) in 2021 were purchased and sorted by screening for freshness to use for research. In the process of homogenization, rotten ones were sorted out. Selected sea buckthorn berries were cleaned with running water and extracted with a juicer (HR1832/00, Philips, Seoul, Korea).

Extracted sea buckthorn juice measuring 50 mL, oligosaccharide 1 Ts (confirmed by several preliminary experiments for the balance between sourness and sweetness), and each gelling agent (guar gum, xanthan gum, and locust bean gum; added 1~9% of extracted sea buckthorn juice) were mixed and stirred at 100 rpm for 3 min in a heating stirrer at 30 °C (MCG05E, Biobridge Co., Hanam, Korea). The mixture in which the gelling agent was dissolved was kept for 1 h in a refrigeration at 4 °C, and they then used as a sample. All these experiments were performed in three runs with all three treatments studied in each run.

### 2.2. Analysis of Fatty Acid Profiles

Extracted sea buckthorn juice measuring 20 mL was saponified with 0.5 N methanolic NaOH (3 mL) at 85 °C for 10 min and cooled to room temperature. After cooling, isooctane (3 mL) and saturated NaCl solution (5 mL) were added to the mixture. The upper isooctane layer containing the FA methyl esters was then collected and passed through an anhydrous Na_2_SO_4_ column. Then, it was analyzed by gas–liquid chromatography using a gas chromatograph (Agilent Technologies 7890-A, Palo Alto, CA, USA). At this point, helium (flow rate of 1 mL min^−1^) was used as the carrier gas.

### 2.3. Analysis of Mineral Profiles

Extracted sea buckthorn juice measuring 0.7 mL, 10 mL of nitric acid, and 3 mL of 30% hydrogen peroxide were mixed. Then, that mixture was digested using 1000 W power at 200 °C. After that, it was cooled at room temperature and quantified by an inductively coupled plasma optical emission spectrometer (ICP-OES; Vista MPX, Varian, Mulgrave, Australia) equipped with a radio frequency source of 40 MHz, charge coupled devices (CCD) simultaneous solid-state detector, peristaltic pump, sea-spray nebulizer connected to cyclonic spray chamber, and high-purity argon (Ar; 99.996% air liquid). The analytical line wavelengths were 317.9 nm of calcium (Ca), 324.8 nm of copper (Cu), 234.4 nm of iron (Fe), 766.5 nm of potassium (K), 280.3 nm of magnesium (Mg), 589 nm of sodium (Na), 213.6 nm of phosphorus (P), 196 nm of selenium (Se), and 206.2 nm of zinc (Zn).

### 2.4. Analysis of Antioxidant Effects

#### 2.4.1. Total Polyphenol Content

Extracted sea buckthorn juice measuring 40 µL and 800 µL of a 10-fold diluted folin ciocalteau reagent was mixed and allowed to stand for 5 min. Then, 800 µL of 7% sodium carbonate aqueous solution (*w*/*v*) was added. After that, the volume in the tube was made up with nano pure water (360 µL) and then allowed to stand for 2 h at room temperature. Then, absorbance was read at 760 nm against the blank using a UV visible spectrophotometer (T60UV, PG Instruments Ltd., Lutterworth, UK).

#### 2.4.2. Total Flavonoid Content

Extracted sea buckthorn juice measuring 0.5 mL was mixed with 1.5 mL of 95% ethanol, 0.1 mL of 10% aluminum chloride hexahydrate, 0.1 mL of 1 M potassium acetate, and 2.8 mL of deionized water (DW). Then, the mixture was incubated at room temperature for 40 min. After that, the absorbance of the reaction mixture was measured at 415 nm against a DW blank on a UV visible spectrophotometer (T60UV, PG Instruments Ltd., Lutterworth, UK).

#### 2.4.3. Superoxide Radical Scavenging Activity

The former comprised a solution of 100 µM xanthine, 60 µM nitro blue tetrazolium in 0.1 M phosphate buffer (pH 7.4), and 0.07 U/mL xanthine oxidase in a total volume of 1 mL. Before the enzyme was added, 0.025 mL of extracted sea buckthorn juice was added. This mixture was incubated at 25 °C for 10 min, and the optical density was read at 560 nm against a blank using a UV visible spectrophotometer (T60UV, PG Instruments Ltd., Lutterworth, UK). The result was presented as IC_50_ (the IC_50_ value represented the concentration of the sea buckthorn berry that caused 50% inhibition).

#### 2.4.4. DPPH Radical Scavenging Ability

Extracted sea buckthorn juice of 0.5 mL, ethanolic stock solution (50 µL) of the antioxidant, and 2 mL of 6 × 10^−5^ M ethanolic solution of 2,20-diphenyl-1-picrylhydrazyl (DPPH) were mixed. The absorbance of this mixture was measured immediately and one hour later at 515 nm by UV visible spectrophotometer (T60UV, PG Instruments Ltd., Lutterworth, UK). The result was presented as IC_50_.

#### 2.4.5. ABTS Radical Scavenging Activity

Extracted sea buckthorn juice measuring 10 µL, 20 µL of myoglobin solution, and 150 µL of 2,2′-azino-bis(3-ethylbenzothiazoline-6-sulfonic acid) (ABTS) reagent (10 mL ABTS and 25 µL 3% H_2_O_2_) were mixed. That mixture was kept at room temperature in the dark, and after 10 min, the absorbance was measured at 405 nm using a UV visible spectrophotometer (T60UV, PG Instruments Ltd., Lutterworth, UK). The result was presented as IC_50_.

#### 2.4.6. Ferric Reducing Antioxidant Power

The ferric reducing ability of plasma (FRAP) reagent was prepared by mixing 25 mL acetate buffer, 2.5 mL 2,4,6-Tris(2-pyridyl)-s-triazine solution, and 2.5 mL ferric chloride solution. Then, 300 µL of that FRAP reagent was warmed at 37 °C, and 10 µL of extracted sea buckthorn juice was added, along with 30 µL DW. Absorbance was measured after 4 min at 593 nm using a UV visible spectrophotometer (T60UV, PG Instruments Ltd., Lutterworth, UK). The result was presented as IC_50_.

#### 2.4.7. Reducing Power

Extracted sea buckthorn juice of 2.5 mL was mixed with 2.5 mL of 200 mM sodium phosphate buffer (pH 6.6) and 2.5 mL of 1% potassium ferricyanide. Then, that was incubated at 50 °C for of 20 min. After, 2.5 mL of 10% trichloroacetic acid was added, and the reaction mixture was centrifuged at 650 rpm for 10 min. The upper layer (2.5 mL) was mixed with 2.5 mL of DW and 0.5 mL of 0.1% ferric chloride. Then, absorbance was measured at 700 nm using a UV visible spectrophotometer (T60UV, PG Instruments Ltd., Lutterworth, UK). The result was presented as IC_50_.

### 2.5. Analysis of Antidiabetes Effects

#### 2.5.1. α–Glucosidase Inhibitory Activity

α-Glucosidase (1 unit/mL) activity inhibition was assayed using 50 μL of extracted sea buckthorn juice incubated with 100 μL of 0.1 M phosphate buffer (pH 7.0) in 96-well plates at 37 °C for 10 min. After pre-incubation, 50 μL of 5 mM p-nitrophenyl-α-d-glucopyranoside solution in 0.1 M phosphate buffer (pH 7.0) was added to each well, and then it incubated at 37 °C for 5 min. The absorbance was measured at 490 nm on a microplate reader (Imark, BioRad, Hercules, CA, USA) before and after incubation. The results were presented as IC_50_ (the amount of sea buckthorn berry can reduce enzyme activity by 50%).

#### 2.5.2. α–Amylase Inhibitory Activity

Extracted sea buckthorn juice of 500 μL was mixed with 500 μL of 0.02 M sodium phosphate buffer (pH 6.9 with 0.006 M sodium chloride) containing 0.5 mg/mL porcine pancreatic α-amylase solution, and it was incubated at 25 °C for 10 min. After the pre-incubation, 500 μL of 1% starch solution in 0.02 M sodium phosphate buffer (pH 6.9 with 0.006 M sodium chloride) was added and then incubated at 25 °C for 10 min. The reaction was stopped by adding 1 mL of 3,5-dinitrosalicylic acid color reagent. The mixture was then incubated in a boiling water bath for 5 min and cooled down to room temperature, and diluted by 10 mL of DW. Absorbance was measured at 540 nm using a UV visible spectrophotometer (T60UV, PG Instruments Ltd., Lutterworth, UK). The results were presented as IC_50_.

### 2.6. Analysis of Texture Properties

Texture profile analysis (TPA) was performed with a texture analyzer (TA-XT Express 20,096, Stable Microsystems Ltd., London, UK). The sample was measured with a two-cycle compression test using a 25 kg load cell. Additionally, a 30 mm diameter cylindrical probe (pre-test speed 3 mm/s, trigger force 5 g, test speed 3 mm/s, return speed 3 mm/s, test distance 5 mm, and time 5 s) was used to compress the sample. This is a method of measuring by pressing a sample between two plates using a compression cylinder, where the TPA recorded the following attributes: hardness, adhesiveness, springiness, chewiness, gumminess, and cohesiveness. In detail, the sample was compressed twice, and hardness was represented by the maximum peak generated in the first compression process, and the cohesiveness of the sample was measured by a force to maintain its original shape. Adhesiveness was measured by the force required for the sample and probe to be separated from each other. Chewiness was measured by the property of making a solid sample swallowable. Gumminess was measured by the property of making a semi-solid sample swallowable. Moreover, springiness was measured by the distance until the probe fell from the sample immediately after compression.

### 2.7. Statistical Analysis

The results of these experiments were tested with a one-way analysis of variance. The test is performed using Scheffe post hoc test to analyze the significant differences between the test groups if the results are significant. Statistical analysis was performed using IBM SPSS statistics (Version 23.0, GraphPad Software Inc., San Diego, CA, USA), and it is determined to be statistically significant if the *p*-value is less than 0.05 (*p* < 0.05).

## 3. Results and Discussion

### 3.1. Fatty Acid Composition

The FA composition of sea buckthorn berry is shown in Table 1. As a result of the analysis, there were 19 different FA. Linoleic acid (18:2n6c; omega-6) is the most abundant FA with a mean value of 33.586 mg/100 g, followed by oleic acid (18:1n9c; omega-9), and linolenic acid (18:3n3; omega-3). On the other hand, there were less saturated FAs: undecanoic acid (11:0), palmitic acid (16:0), and stearic acid (18:0). Although the main type of fat in fruit is healthy unsaturated fat, there are also some saturated fats. Whether saturated fat in fruits is as bad for our health as fat in animal products is still a big controversy, but limiting the amount people consume is still recommended. Considering this, the results of this study show that sea buckthorn berries can be a good choice for the elderly because of their low saturated FA. According to previous studies, the intake of saturated FA should be avoided in a balanced diet because the excessive intake of fat is related to a high prevalence of myocardial infarction, hyper-cholesterolemia, increased low-density lipoprotein (LDL) cholesterol and blood pressure, and other chronic disease [6]. Thus, reducing saturated FA intake and increasing unsaturated FA intake are recommended. Therefore, sea buckthorn berries can be a great selection for the elderly because of the high unsaturated FA associated with reducing the prevalence of cancer, dementia, and chronic diseases such as cardiovascular disease and autoimmune diseases. In particular, linoleic acid, the unsaturated FA most abundant in sea buckthorn berries, is well known to be associated with a decrease in LDL cholesterol and an increase in high-density lipoprotein cholesterol. Moreover, oleic acid and linolenic acid from sea buckthorn berries maintain a healthy cell membrane, provide energy, and provide vitamin E, which is a powerful antioxidant [7].

Relatively, sea buckthorn berry had a higher value of unsaturated FA (71.951 mg/100 g) than celastrus berry (25.2 mg/100 g), a natural antioxidative food that is recognized good for the health of the elderly [8]. According to this study, sea buckthorn berries are superior to celastrus berries in the increase in unsaturated FA, which performs an important role in the human body, such as controlling blood cholesterol levels, relieving inflammation, and stabilizing heart rhythm and various biological reactions.

### 3.2. Mineral Composition

The mineral composition of sea buckthorn berry is shown in Table 2. As a result of the analysis, there were nine different minerals. K is the most abundant mineral with a mean value of 600.918 mg/100 g, followed by Na, Ca, P, Fe, Mg, Zn, Mn, and Cu. Sea buckthorn berries had relatively higher values for K, Na, Ca, P, Fe, Mg, Zn, Mn, and Cu than strawberries, raspberries, blueberries, black currents, blackberries, and gooseberries, which are natural antioxidative fruits recognized to be beneficial for the elderly [9]. This shows that sea buckthorn berry is an appropriate food for increasing mineral intake and performs an important role in preserving cellular water balance and helping to improve protein metabolism.

In particular, the mean value of K in sea buckthorn berry was three to four times higher than strawberry (153 mg), orange (181 mg), and grape fruit (135 mg) known as good mineral sources [10]. This shows that sea buckthorn berry is a suitable food for increasing K intake, one of the most important minerals in the body, and can help regulate ecological balance, muscle contraction, and nerve signals in the body [11]. The mean value of Na of sea buckthorn berry (388.066 mg) was also significantly higher. Similarly, the mean value of Ca in sea buckthorn berry (169.546 mg) was four to ten times higher than strawberry (16 mg), orange (40 mg), and grape fruit (22 mg), which helped protect bones and teeth, as well as enzyme production and hormone distribution. The mean value of P in sea buckthorn berry (80.213 mg), an important constituent of nucleic acids and cell membranes, was also three to five times higher than strawberry (24 mg), orange (14 mg), and grape fruit (18 mg), and lack of P intake leads to bone mineralization. In addition, the mean value of Fe in sea buckthorn berry (31.877 mg), an important component in the cellular respirator, is significantly higher than strawberry (0.41 mg), orange (0.1 mg), and grape fruit (0.08 mg). On the other hand, there are also side effects of extreme addiction, for which its essential consequences involve slow bone growth and increasing death. Therefore, the requirement for daily Fe intake is strictly limited to 0.3~170 mg/100 g, which depends on the variation in the population. Fortunately, however, in this study, the Fe content of sea buckthorn berry was detected within the limit, which is appropriate for human intake [12]. In the case of Mg, the mean value of Mg in sea buckthorn berry was 24.310 mg/100 g, more than twice as high as strawberry (13 mg), orange (10 mg), and grape fruit (9 mg). As a result, sea buckthorn berry is a suitable food for increasing Mg intake, improving enzymatic function, and maintaining nerve electrical ability. The Zn value of sea buckthorn berry (2.255 mg) is also significantly higher than that of strawberry (0.14 mg), orange (0.07 mg), and grape fruit (0.07 mg), which helps improve immune function and is significant for proper perception of taste and smell. Moreover Mn, similarly to other minerals, shows a high mean value in sea buckthorn berry (2.039 mg). Finally, the mean value of Cu was found to be 1.211 mg, an element of various enzymes and an important factor in numerous physiological functions.

According to this study, sea buckthorn berry is an excellent food that increases the intake of minerals and helps vital functions in the human body, such as many enzymatic reactions, energy production, neurostimulatory transmission, and multiple biological reactions [11].

### 3.3. Antioxidant Effects

#### 3.3.1. Total Polyphenol Content

As a result of this study, the total polyphenol content in sea buckthorn berry was 145.566 g GAE/100 g (Table 3), which was higher than that of red currant (105 g GAE/100 g), one of the representative antioxidant foods [13]. As such, sea buckthorn berry contains a large amount of polyphenol, which is expected to have many potential health benefits. Polyphenol contained in sea buckthorn berry is a powerful antioxidant and has been reported to exhibit antibacterial, antiviral, anticancer, anti-inflammatory, and vasodilation [14]. Recently, interest in plant-derived antioxidants has increasing because it may replace synthetic food antioxidants. Sea buckthorn berry may be an appropriate replacement for polyphenolic compounds, which may have a particular significant impact on the health of the elderly.

#### 3.3.2. Total Flavonoid Content

Further analysis of sea buckthorn berry showed that the total flavonoid content (12.375 mg RE/100 g) was higher than broccoli (9.4 mg/100 g), apple (9.4 mg/100 g), and black grape (3.0 mg/100 g), which are great sources of flavonoid known to be beneficial to the elderly (Table 3). Therefore, high total flavonoids may help inhibit lipid peroxidation, free radical scavenging, metal chelates, and anti-inflammatory activity [15]. In addition, it can reduce the risk of developing and progressing atherosclerosis and insulin resistance, and it can help lower blood pressure.

#### 3.3.3. Superoxide Radical Scavenging Activity

As shown in Table 3, the superoxide radical scavenging activity (SOD) IC_50_ of sea buckthorn berry is 211.287 μg/mL. Previous studies have shown that superoxide is a biologically important entity because it can be decomposed into powerful oxidative species such as singlet oxygen or hydroxyl radicals and is very harmful to the cellular components of the biological system [16]. Moreover, the anionic superoxide radical (O^2^^•−^) is evolved by a membrane-binding enzyme, nicotinamide adenine dinucleotide phosphate oxidase, through the reduction of one electron from free molecular oxygen, which in cells turns into destructive active oxygen such as hydroxyl radicals and hydrogen peroxide. Thus, the removal of peroxide radical anions produced by this enzymatic pathway will be valuable in solving various health problems. Therefore, this result suggests that sea buckthorn berry is an adequate antioxidative food for the elderly. In addition, results similar to sohiong (225.73 μg/mL), which is known to have high antioxidant properties among wild edible fruits [17].

#### 3.3.4. DPPH Radical Scavenging Activity

The DPPH free radical scavenging activity IC_50_ was 48.755 μg/mL (Table 3), close to standard ascorbic acid (40.16 μg/mL), showing influential antioxidant properties. In addition, the results were higher than blackberries (16.84 μg/mL) and blueberries (9.82 μg/mL), known as representative antioxidants [18]. Previous studies have reported that intake of antioxidant-containing foods such as sea buckthorn berry is beneficial to human health since they can protect the human body from harmful free radicals and inhibit the progression of many chronic diseases [16]. Therefore, this result suggests that sea buckthorn berry can help reduce the danger of cancer, cardiovascular diseases, and so on.

#### 3.3.5. ABTS Radical Scavenging Activity

As a result of this study (Table 3), the ABTS radical scavenging activity IC_50_ of sea buckthorn berry (9.539 μg/mL) was higher than that of cucumber (6.71 μg/mL) and snake gourd (6.93 μg/mL), which are good antioxidant sources recognized to be beneficial to the elderly. This result is closely related to the high concentration of phenol. In other words, sea buckthorn berry is expected to exhibit high antioxidant activity due to the existence of phenol.

#### 3.3.6. Ferric Reducing Antioxidant Power

The antioxidant efficiency of sea buckthorn berry determined by the current analysis of FRAP assay depends on the redox potential of the compound under being studied characterized by molecular complexity, with ferric reducing antioxidant power (FRAP) IC_50_ of sea buckthorn berry (61.107 μg/mL) higher (Table 3) than standard ascorbic acid (40.16 μg/mL), showing good FRAP antioxidant effects [19]. In addition, the results were similar with strawberries (61.73 μg/mL) and raspberries (61.94 μg/mL), known as representative antioxidants [20]. This result is also closely related to the high concentration of phenol. In other words, sea buckthorn berry is expected to exhibit excellent antioxidant activity due to the existence of phenol.

#### 3.3.7. Reducing Power

Reducing power is also related to the polyphenol content, and is one of the widely used methods for evaluating the antioxidant activity of food. That is, the reducing power is generally associated with the presence of reducing agents, which act as antioxidants by breaking free radical chains by donating hydrogen atoms [21]. Therefore, this research estimated the iron reducing capacity of sea buckthorn berry from the ability to decrease the Fe^3+^-ferricyanide complex to iron form by providing electrons. As a result, the reducing power IC_50_ of sea buckthorn berry was 15.095 μg/mL (Table 3). In addition, the result was somewhat higher than that of ginseng berries (11 μg/mL), known as representative antioxidant foods [22]. Here, we expected that this antioxidant activity of sea buckthorn berry is likely due to the presence of polyphenols that can act as free radicals scavengers by donating electrons or hydrogen.

### 3.4. Antidiabetes Effects

Sea buckthorn berry was evaluated for its antidiabetic properties by the inhibition of α-amylase and α-glucosidase enzymes. α-amylase and α-glucosidase are responsible for carbohydrate digestion, and the inhibition of these enzymes can decrease blood sugar levels immediately after meals by reducing the breakdown of polysaccharides into glucose [23]. In this study, the IC_50_ values of sea buckthorn berry for α-glucosidase and α-amylase inhibition were 0.972 mg/mL and 0.745 mg/mL. In addition, compared to date (α-glucosidase inhibition 0.35 mg/mL and α-amylase inhibition 0.78 mg/mL), which is known to have antidiabetic activity, the antidiabetic activity of sea buckthorn berry was found to be higher or similar [24]. Previous studies have shown that anti-diabetic activity is associated with phenolic compounds and flavonoids, and polyphenols in fruits and vegetables in particular increase antidiabetic activity [25]. In addition, biological activity due to polyphenols in many plant species has been shown to effectively decrease early diabetes-related enzyme activity. Therefore, we assume that the α-glucosidase and α-amylase inhibition activity of sea buckthorn berry is likely due to the presence of polyphenols.

### 3.5. Hardness of Sea Buckthorn Jelly

One of the principal quality parameters that determine the sensory features of food for the elderly is texture. Moreover, since Korea has become a super-aged society, expanding the consumer industry market for elder-friendly foods is suggested, which can play a vital role in the elderly in which balanced food intake is particularly important. Therefore, the Korea Ministry of Food and Rural Affairs quantified the chewing hardness standard of elder-friendly foods to three levels (the hardness that can be chewed using teeth, gum, and tongue) of KS [4]. In addition, in the case of the third stage, which cannot be consumed by teeth or gums, the viscosity standard that the elderly can swallow was also presented. However, despite the recent need for the development of elder-friendly foods in the food industry market, the relevant basic data are very limited in the literature. For example, because fruit has a tough skin and flesh relatively firm due to its fiber, research on various textural qualities is essential for enabling the elderly to ideally consume fruit as an antioxidant food resources. In addition, extracted fruit juice, which can be generally easily consumed, is not viscous, and there is a risk of dysphagia in the elderly who are weak in the mouth or esophageal muscles. Hence, this study attempted to develop fruit jelly that can be consumed according to each oral condition of the elderly classified by KS by adding a gelling agent to fruit juice. That is, this research was conducted to study the effect on the textural quality of sea buckthorn jelly according to the gelling agents (guar gum, xanthan gum, and locust bean gum) as a naturally derived food additive. The results are as follows.

Table 4 presents the characteristics of each texture parameter of the sea buckthorn jelly to which various gelling agents are added, respectively. In the case of the first level for KS, samples with 7 to 9% guar gum added had a hardness (55,000~500,000 N/m^2^) that could be chewed with the teeth of the elderly, and samples with more than 9% guar gum added had a hardness (22,000~50,000 N/m^2^) that could be chewed with the tongue. In the second level for KS, 5% guar gum was added to the sample, which could be chewed with the gum of the elderly, while more than 9% guar gum was still chewable with the tongue. Moreover, the sample that added 7 to 9% of locust bean gum had a hardness that could be chewed with gum. In the third level for KS, samples with up to 3% guar gum added had a hardness (~20,000 N/m^2^) that could be chewed with the tongue of the elderly, and samples with up to 9% xanthan gum added had a hardness that could be chewed with the tongue. Samples with up to 5% locust bean gum added had a tongue-chewable hardness, while samples with a viscosity of 336.800 mPa.s did not form any form of less than 3% of locust bean gum. Other texture parameters (adhesiveness, springiness, chewiness, gumminess, cohesiveness, and resilience) also showed similar results. However, among the texture parameters, hardness is a major texture attribute that plays an important role in the food preference of the elderly. Therefore, KS was established based on the hardness of food, and this study also attempted to confirm the applicability of the gelling agent based on the hardness of sea buckthorn jelly.

Based on these results, the linear regression equation evaluated for guar gum is ‘Y = 884,208X − 6597.2’ (where X is the concentration of guar gum, and Y is the hardness of sea buckthorn jelly). Moreover, the linear regression equations evaluated for each xanthan gum and locust bean gum are ‘Y = 721,413X − 21,412’ (where X is the concentration of xanthan gum and Y is the hardness of sea buckthorn jelly) and ‘Y = 192,070X − 1863.5’ (where X is the concentration of locust bean gum and Y is the hardness of sea buckthorn jelly). As can be seen from Figure 1, 3.008% of guar gum, 5.740% of xanthan gum, and 11.383% of locust bean gum have hardness that can be chewed with the tongue of the elderly. In addition, at guar gum 3.234 to 6.401%, xanthan gum 6.018 to 9.900% and locust bean gum 12.424 to 27.002% are added, and sea buckthorn jelly has a hardness that can be chewed with gum of the elderly. Moreover, 6.966% of guar gum, 10.592% of xanthan gum, and 29.606% of locust bean gum have a level of hardness that can be chewed with teeth.

The above results proved that only a small amount of guar gum and xanthan gum can provide meaningful development in the hardness of liquid food. In addition, xanthan gum not only helps suspension of solid particles such as spices in addition to liquid food but also helps thicken commercial egg replacements made from egg whites to replace synthetic emulsifiers, according to a prior study [26]. Therefore, xanthan gum and guar gum are applied to a variety foods such as beverages, sauces, dressings, dairy products, meat and poultry products, bakery products, and confectionery products. Moreover, these gelling agents do not significantly alter the food’s natural color or taste, so it is one of the ways to increase the viscosity of sap preferred by patients who are suffering from swallowing difficulties in hospitals. Moreover, the optimization of elder-friendly foods through various textural control such as hardness and viscosity helps elderly customers who are difficult to chew and swallow easily consume various fruits, and considering the current situation in the super-aged society, it is possible to positively expand the outlook of the elderly food industry. the guar gum used in this study is a galactomannan polysaccharide derived from guar beans, which is known to lower serum cholesterol and lower blood glucose levels, and it is widely used in foods such as xanthan gum and guar gum discussed earlier. It is also very cost effective because it has almost eight times more water-thickening capability than other gelling agents such as cornstarch and requires only a very little amount to produce an adequate viscosity. In other words, since it is less necessary than other gelling agents, there is an effect of reducing costs. In addition, in addition to the influence of locust bean gum on viscosity, high fluidity and deformability provide favorable rheological properties for elder-friendly foods [27]. In one previous study, locust bean gum, a galactomannan vegetable gum derived from the seeds of a carob tree, was used as a food thickener, which could help lower blood sugar and blood fat levels [28]. It is also used in infant formula to reduce reflux from the stomach to the esophagus.

Comparing these three gelling agents, the guar gum was dissolved more than the locust bean gum and xanthan gum because of the extra galactose branches. Moreover, when the same amount (more than 3% of the concentration of the gelling agent) is added, guar gum shows the highest hardness in the sea buckthorn jelly, whereas xanthan gum shows the lowest hardness in the sea buckthorn jelly. By examining these results, this was confirmed as well as the possibility of developing antioxidant foods for the health of the elderly. When the concentration of the gelling agents is less than 3%, considering that the sample with locust bean gum has not solidified, it would be meaningful to develop viscous juice to prevent dysphagia in the elderly. Meanwhile, the viscosity of the food varies depending on the concentration, pH, and particle size of the gelling agent, stirring time, temperature, speed, and so on. Therefore, further research on the elder-friendly food industry for the elderly who are difficult to chew and swallow in the future is continuously needed.

## 4. Conclusions

Gelling agents are known to help the elderly eat food easily by providing a hardness suitable for each possible mastication stage. Changing the textural quality of these foods is one of the most common applications used to supplement the disproportionate nutrition of the elderly. Here, the components used to characterize food include size, color, shape, taste, viscosity, and texture. Based on this information, the results of the study suggest that controlling the hardness of sea buckthorn jelly with a gelling agent can help easily suffocating older people eat variety of foods. Moreover, considering that a moderately viscous diet is commonly used in the elderly with dysphagia, the results of this study confirm that the use of gelling agents in foods may be linked to a reduction in pharyngeal residues and may also help reduce aspiration pneumonia in the elderly with dysphagia. In other words, by establishing the amount of each gelling agent suitable for the hardness of each stage of KS, it is possible to provide food with an appropriate hardness to the elderly with different chewing and swallowing abilities. Therefore, it is considered to be a significant study for balanced nutrition intake for the elderly who have not been sufficiently nourished due to deterioration of their intake ability due to aging. These studies can be used as very meaningful information in Korea, which has entered a super-aged society. With the increase in the elderly population, the elderly has become consumers who have a great influence on the consumption economy and pursue a healthy well-being life. Accordingly, the demand for elder-friendly food is increasing significantly, and the food industry is also changing to reflect this. Therefore, this study shows the possibility of using sea buckthorn berry with excellent beneficial biological activity such as essential fatty acids, minerals, and antioxidants, and the progress of modern research on the formulation design of health functional foods by focusing on food texture using gelling agents. Unfortunately, however, the study has some limitations. Above all, this study did not conduct sensory test on the elderly, who are actual elder-friendly food consumers, and it seems necessary to check various preferences of sea buckthorn jelly for the elderly with different chewing and swallowing abilities in the future.

In conclusion, by adding gelling agents to food at an appropriate concentration, the risk of choking as well as nutritional imbalance can be improved by controlling the hardness and viscosity of food consumed by the elderly who are prone to suffocation. This research study could help enlighten the food industry about the usefulness of gelling agents as a texture modifier in a special diet for the elderly. In other words, the easy intake of the elderly should be an essential condition for the enjoyment of eating; thus, applying gelling agents to food to induce various textures can lead to an improvement in the quality of life of the elderly. Understanding work standards linked to chewing and swallowing capabilities of the elderly and hardness and viscosity is important for the elder-friendly food industry not only in technical features but also in commercial features; thus, this study can provide motivation for the product expansion of elder-friendly foods overall. That is to say that, as the elder-friendly food industry is still in its early stages, such research can be used as basic data for industrial development; therefore, many such studies should be conducted. Only when these studies are conducted in various ways can the elder-friendly food industry have high growth and high-quality results. On the other hand, the sea buckthorn jelly for the elderly developed in this study has limitations in that no sensory evaluation has been conducted on the elderly, who are actual consumers. Therefore, further research should be conducted in this regard. Furthermore, it is necessary to further subdivide and define the hardness to identify and accommodate elderly consumers with different intake capacities in more subdivisions than now. Therefore, further research is needed to introduce improvement guidelines to promote its industrial utilization.

## Figures and Tables

**Figure 1 foods-11-01892-f001:**
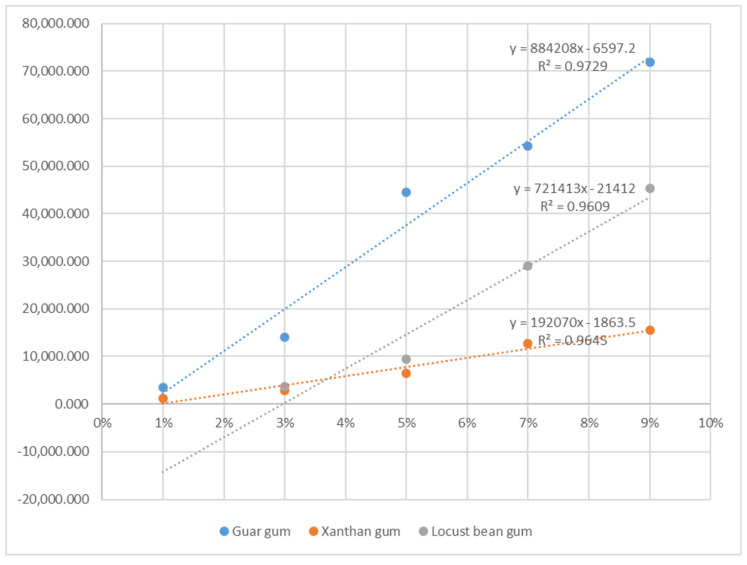
Regression equation for hardness of sea buckthorn jelly added with gelling agent. The y-scale is “hardness (N/m^2^)”, and the x-scale is “gelling agent concentration (%)”.

**Table 1 foods-11-01892-t001:** Analyzed fatty acid composition of sea buckthorn berry.

Composition	Content (mg/100 g)
Saturated fatty acid	28.049 ± 2.983
C_(11:0)_	13.126 ± 0.666
C_(14:0)_	0.131 ± 0.006
C_(15:0)_	0.117 ± 0.004
C_(16:0)_	10.233 ± 0.501
C_(17:0)_	0.067 ± 0.003
C_(18:0)_	4.148 ± 0.204
C_(21:0)_	0.083 ± 0.001
C_(23:0)_	0.080 ± 0.004
C_(24:0)_	0.064 ± 0.002
Unsaturated fatty acid	71.951 ± 1.967
C_(18:1t)_	0.226 ± 0.013
C_(18:1n9c)_	18.911 ± 0.936
C_(18:1n7c)_	0.525 ± 0.020
C_(18:2t)_	0.127 ± 0.003
C_(18:2n6c)_	33.586 ± 1.678
C_(18:3t)_	0.137 ± 0.005
C_(18:3n3)_	17.681 ± 0.878
C_(20:1)_	0.205 ± 0.010
C_(22:1)_	0.133 ± 0.005
C_(24:1)_	0.420 ± 0.011

Mean ± S.D.

**Table 2 foods-11-01892-t002:** Analyzed mineral composition of sea buckthorn berry.

Composition	Content (mg/100 g)
Calcium	169.546 ± 8.334
Iron	31.877 ± 1.435
Magnesium	24.310 ± 1.101
Phosphorous	80.213 ± 3.016
Potassium	600.918 ± 27.887
Sodium	388.066 ± 17.043
Zinc	2.255 ± 0.017
Copper	1.211 ± 0.050
Manganese	2.039 ± 0.201

Mean ± S.D.

**Table 3 foods-11-01892-t003:** Analyzed antioxidant and antidiabetic effects of sea buckthorn berry.

		Status
Antioxidation	Total polyphenol content (g GAE/100 g)	145.566 ± 4.962
	Total flavonoid content (mg RE/100 g)	12.375 ± 1.382
	Superoxide radical scavenging activity IC_50_ (μg/mL)	211.287 ± 10.844
	DPPH radical scavenging activity IC_50_ (μg/mL)	48.755 ± 3.000
	ABTS radical scavenging activity IC_50_ (μg/mL)	9.539 ± 0.071
	Ferric reducing antioxidant power IC_50_ (μg/mL)	61.107 ± 0.792
	Reducing power IC_50_ (μg/mL)	15.095 ± 0.660
Antidiabetes	α-glucosidase inhibitory activity IC_50_ (mg/mL)	0.972 ± 0.032
	α-amylase inhibitory activity IC_50_ (mg/mL)	0.745 ± 0.024

Mean ± S.D.

**Table 4 foods-11-01892-t004:** Texture of sea buckthorn jelly added with gelling agent.

Parameters	Concentration of Gelling Agent (%)	Gelling Agent			F-Value (*p*-Value)
		Guar Gum	Xanthan Gum	Locust Bean Gum	
Hardness (N/m^2^)	1	3548.000 ± 170.420	1194.333 ± 50.589	N.D. (Viscosity (mPa.s) 336.800 ± 14.681)	11,593.671 *** (0.000)
	3	13,953.667 ± 664.290	2878.667 ± 144.922	3730.000 ± 72.459	729.949 *** (0.000)
	5	44,479.000 ± 1284.075	6417.000 ± 298.679	9480.667 ± 245.082	2233.602 *** (0.000)
	7	54,280.000 ± 990.625	12,738.667 ± 204.534	28,940.333 ± 1240.829	1541.114 *** (0.000)
	9	71,805.667 ± 3142.662	15,471.333 ± 761.105	45,338.000 ± 2019.845	493.037 *** (0.000)
Adhesiveness (N·s/m^2^)	1	−1.663 ± 0.026	−1.762 ± 0.056	N.D.	2342.922 *** (0.000)
	3	−1.383 ± 0.034	−1.350 ± 0.040	−2.124 ± 0.068	234.536 *** (0.000)
	5	−0.436 ± 0.019	−0.312 ± 0.015	−1.929 ± 0.091	829.669 *** (0.000)
	7	−0.244 ± 0.011	−0.139 ± 0.007	−1.109 ± 0.047	1130.750 *** (0.000)
	9	−0.163 ± 0.007	−0.086 ± 0.004	−0.253 ± 0.007	536.669 *** (0.000)
Springiness (mm)	1	0.657 ± 0.032	0.680 ± 0.026	N.D.	773.788 *** (0.000)
	3	0.900 ± 0.040	0.737 ± 0.035	0.460 ± 0.020	137.701 *** (0.000)
	5	0.990 ± 0.045	0.767 ± 0.035	0.643 ± 0.029	128.262 *** (0.000)
	7	1.310 ± 0.050	0.927 ± 0.006	0.817 ± 0.038	152.176 *** (0.000)
	9	2.253 ± 0.112	0.930 ± 0.018	0.973 ± 0.006	392.135 *** (0.000)
Chewiness (N/m^2^)	1	308.900 ± 15.342	238.333 ± 7.086	N.D.	4694.143 *** (0.000)
	3	851.500 ± 38.431	327.000 ± 15.519	348.267 ± 5.008	455.105 *** (0.000)
	5	1528.967 ± 55.853	455.700 ± 10.958	830.667 ± 11.172	793.636 *** (0.000)
	7	8013.067 ± 371.071	578.867 ± 23.412	1066.000 ± 34.028	1116.567 *** (0.000)
	9	9757.100 ± 270.363	632.900 ± 27.022	1288.633 ± 37.335	3098.811 *** (0.000)
Gumminess (N/m^2^)	1	310.800 ± 15.451	102.867 ± 5.046	N.D.	8857.172 *** (0.000)
	3	960.067 ± 29.122	146.300 ± 5.957	351.800 ± 4.161	1788.929 *** (0.000)
	5	2048.567 ± 111.261	256.200 ± 5.803	852.800 ± 9.506	599.656 *** (0.000)
	7	3927.367 ± 172.903	832.333 ± 37.239	1614.267 ± 18.657	737.005 *** (0.000)
	9	6170.933 ± 309.331	1170.067 ± 53.603	2486.100 ± 84.926	571.778 *** (0.000)
Cohesiveness	1	0.593 ± 0.021	0.737 ± 0.032	N.D.	936.068 *** (0.000)
	3	0.737 ± 0.012	0.743 ± 0.035	0.547 ± 0.025	56.117 *** (0.000)
	5	0.750 ± 0.010	0.753 ± 0.035	0.637 ± 0.021	22.472 *** (0.002)
	7	0.797 ± 0.031	0.877 ± 0.023	0.687 ± 0.006	54.600 *** (0.000)
	9	0.860 ± 0.042	0.923 ± 0.006	0.890 ± 0.010	38.714 *** (0.000)
Resilience	1	0.097 ± 0.006	0.217 ± 0.006	N.D.	1590.500 *** (0.000)
	3	0.130 ± 0.005	0.223 ± 0.006	0.090 ± 0.000	316.000 *** (0.000)
	5	0.240 ± 0.010	0.250 ± 0.010	0.097 ± 0.006	283.857 *** (0.000)
	7	0.483 ± 0.021	0.360 ± 0.017	0.120 ± 0.001	417.942 *** (0.000)
	9	0.520 ± 0.026	0.457 ± 0.021	0.123 ± 0.006	349.841 *** (0.000)

N.D. means not detected. *** significant at *p* < 0.001, respectively.

## Data Availability

The data presented in this study are available on request from the corresponding author.

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
