# Peer review of "Texture Characteristics of Sea Buckthorn (Hippophae rhamnoides) Jelly for the Elderly Based on the Gelling Agent"

_foods, 2022, doi:10.3390/foods11131892_

Round 1

Reviewer 1 Report

Texture characteristics of sea buckthorn jelly for the elderly based on the gelling agent

Manuscript is well organized and its aim is very surprising for me: to design jelly based on sea buckthorn juice and thickened to elderly to be acceptable by chewing. There are defined three levels of Korean Industrial Standards (KS) for hardness: chewed by teeth, chewed by gum and chewed by tongue. Each level has defined hardness range in N/m^2.

General comments:

- Tables and figures can be placed directly in manuscript as close as possible to their comments, why this manuscript has all tables and figure at the manuscript end?

- Why I received manuscript without line numbers?

- Why thickening of the juice is done? Elderly can drink juice without problem, I think.

 Other comments are given in the following list.

 -        Part 2.1 Sample preparation – second paragraph is not clearly described in part how gelling agents were added, mixed at which temperature, how intensive was mixing etc.

-        Part 2.6 Analysis of Texture Properties – there is need to provide figure of sample and TA-XT tool (was that sample for compression cylinder compressed between two plates or penetration of cylinder into the large cylinder shape sample?)

-        There is need also to provide figure how harness and other parameters were evaluated from compression deformation – force data!

-        Part 3.3.3 Superoxide Radical Scavenging Activity: first sentence contains word Table 3 twice. Purge second one and at first Table 3 use “T” instead “t”.

-        Prepared jellies fits the KS but why did not test these jellies on humans? Sensory evaluation is good tool to verify instrumental measurement.

-        Table 4 given data is provided on miliNewtons per squared meters. Standard deviation has order hundreds or thousands of Newtons per squared meters. It has no sense. Please, give numbers in Newton units per square meter at mean values and standard error.

-        Figure 1 describe scales: y scale “Hardness” in correct units (N/m^2), x scale  “gelling agent concentration (%).

Author Response

I attached the file for the detailed answer. I am sorry for the lack of it, and I revised it by reflecting the comments you pointed out as much as possible. Thank you very much for your kind detailed review despite your busy schedule.

Reviewer 2 Report

This proposed study “Texture characteristics of sea buckthorn (Hippophae rhamnoides) jelly for the elderly based on the gelling agent” is well written and designed, and it need the major revision according to  the following comments:

Comment 1: Provide the information regarding the gelling agent in the introduction part?

Comment 2: Authors more identify the problems and what is novelty of this study? Revise it in more comprehensive way

Comment 3: What is consumption of sea buckthorn berries?

Comment 4: Is it also in the market in the form of product, if yes than why this study was necessary? 

Comment 5: Why the 1 to 9% percentage was selected?

Comment 6: Section 2.4.4: “the absorbance of this mixture is” replace “the absorbance of this mixture was”

Comment 7: Section 3.1: is it enough that sea buckthorn berries are low in SFA for elderly people?

Comment 8: There are number of berries available, which are low in SFA and chewing abilities are better.

Comment 9: There is some minor English mistake must be identified.

Comment 10: Section 3.3: Authors should add the comparative composition of some references berries, which have the similar characteristics or lower than sea buckthorn berries.

Comment 11: Provide the practical significance of this study in conclusion part

Author Response

(The authors gave the same response as above.)

Round 2

Reviewer 2 Report

All the comments are well addressed